# *Tenebrio molitor* Larva Trypsin Hydrolysate Ameliorates Atopic Dermatitis in C57BL/6 Mice by Targeting the TLR-Mediated MyD88-Dependent MAPK Signaling Pathway

**DOI:** 10.3390/nu15010093

**Published:** 2022-12-24

**Authors:** Meiqi Fan, Nishala Erandi Wedamulla, Young-Jin Choi, Qun Zhang, Sung Mun Bae, Eun-Kyung Kim

**Affiliations:** 1Division of Food Bioscience, College of Biomedical and Health Sciences, Konkuk University, Chungju 27478, Republic of Korea; 2Department of Food Science and Nutrition, College of Health Science, Dong-A University, Busan 49315, Republic of Korea; 3Center for Silver-Targeted Biomaterials, Brain Busan 21 Plus Program, Dong-A University, Busan 49315, Republic of Korea; 4Department of Health Sciences, The Graduate School of Dong-A University, Busan 49315, Republic of Korea; 5Department of Export Agriculture, Faculty of Animal Science and Export Agriculture, Uva Wellassa University, Badulla 90000, Sri Lanka; 6Gyeongnam Agricultural Research and Extension Services, Jinju 52733, Republic of Korea; 7Center for Food & Bio Innovation, Dong-A University, Busan 49315, Republic of Korea

**Keywords:** *Tenebrio molitor*, atopic dermatitis, MAPK, trypsin hydrolysate

## Abstract

Atopic dermatitis (AD) is a widely researched chronic inflammatory skin disease with a complex etiology. The increased prevalence of AD necessitates exploration of natural sources as potential therapeutic agents with limited side effects. In the current study, a 1-chloro-2,4-dinitrobenzene (DNCB)-induced AD mouse model was used to examine the anti-AD effects of *Tenebrio molitor* trypsin hydrolysate (TMTH) and its underlying molecular mechanism. DNCB-treated mice were treated with TMTH (1 and 10 mg/kg), and prednisolone (3 mg/kg) was used as the positive control. Serum and skin tissue samples were collected for subsequent analyses. The expression levels of proteins linked to the myeloid differentiation primary response 88 (MyD88)-dependent mitogen-activated protein kinase (MAPK) signaling pathway and serum IgE levels were estimated via Western blotting technique and ELISA (enzyme-linked immunosorbent assay), respectively. Inflammatory cell infiltration and thickening of the dorsal skin were measured using toluidine blue and hematoxylin and eosin staining, respectively. Oral administration of TMTH significantly reduced mast cell infiltration and dermal and epidermal thickness. Moreover, TMTH treatment reduced serum IgE levels. Western blotting confirmed that TMTH treatment suppressed the MyD88-dependent MAPK signaling pathway. Therefore, TMTH substantially inhibited AD-like skin lesion formation via immunomodulation, showing considerable potential for AD treatment.

## 1. Introduction

Eczema, widely known as atopic dermatitis (AD), is a chronic inflammatory skin disease associated with several symptoms, including swelling, cracked skin, erythema, and pruritus [1,2]. This chronic, recurrent, and hereditary allergic skin condition has become a major global health concern owing to its increasing prevalence worldwide [1]. The pathophysiology of AD has not been fully explained. However, several studies have suggested that immunological dysregulation along with dysregulation of the epidermal barrier plays a key role in the initiation and aggravation of AD [3]. 

The role of Toll-like receptors (TLRs) in the pathogenesis of various inflammatory diseases is well documented. Stimulation of TLRs activates a downstream signaling cascade that activates the transcription factor nuclear factor (NF)-κB. Activated NF-κB induces the release of proinflammatory cytokines, such as interleukin (IL)-6, IL-1β, and tumor necrosis factor (TNF)-α. Activation of TLRs leads to interactions with the adaptor protein myeloid differentiation primary response 88 (MyD88) via Toll-interleukin-1 receptor-domain interactions. Subsequently, downstream proteins are recruited by these adaptor proteins in the signaling cascade to contribute to signal amplification by the interleukin receptor-associated kinase (IRAK) family. This interaction induces the phosphorylation of IRAK4, followed by the phosphorylation of IRAK1. IRAK1 associates with and activates TNF receptor-associated factor 6 (TRAF6). The IRAK1/TRAF6 complex dissociates from the intracellular receptor complex and initiates the formation of a novel complex with the E2 ligases Ubc13 and Uev1A. This complex then associates with and activates different members of the mitogen-activated protein kinase (MAPK) family, such as the extracellular signal-regulated kinase (ERK), c-Jun *N*-terminal kinase (JNK), and p38 [4,5].

Protein shortage has become a global problem owing to the increasing human population. This has led to the exploration of novel protein sources with various functional and bioactive properties. Edible insects have recently been identified as promising alternative sources of proteins with various functional properties. Accordingly, the functional properties of *Tenebrio molitor* have been reported [6,7]. Studies assessing the bioactive characteristics of proteins derived from *T. molitor* have found that their functional properties can be significantly altered by extraction conditions [8]. Bioactive peptides obtained via enzymatic hydrolysis exhibit beneficial in vivo physiological effects, including enhanced digestibility and bioavailability [9]. Several studies have demonstrated various bioactive properties of *T. molitor* protein hydrolysates. Tang et al. have shown enhanced antioxidant activities and significant changes in the amino acid composition of *T. molitor* protein hydrolysate compared to the water extract [10]. Other studies have demonstrated promising antioxidant properties of the protein hydrolysate of *T. molitor* larvae [9]. Additionally, many studies have reported the antidiabetic [11] and hepatoprotective [12] effects of *T. molitor* (mealworm) protein hydrolysates. 

Although research has proven the bioactive properties of protein hydrolysates of *T. molitor* larvae, studies exploring their effect on AD via the TLR-mediated MyD88-dependent MAPK signaling pathway remain scarce. Thus, the current study assessed the effect of *T. molitor* larva trypsin hydrolysate (TMTH), a 1–10 kDa molecular fraction, on AD using 1-chloro-2,4-dinitrobenzene (DNCB)-treated experimental mice. The effect of TMTH on AD was assessed based on the expression levels of different cytokines and histopathological changes. Our findings revealed that TMTH treatment effectively reduced the expression levels of IRAK4, MyD88, and TRAF6 proteins, along with those of *p*-p38, *p*-ERK, and *p*-JNK, which are associated with the MAPK signaling pathway. Overall, TMTH ameliorated AD via regulation of the TLR-mediated MyD88-dependent MAPK signaling pathway.

## 2. Materials and Methods

### 2.1. Materials

TMTH was supplied by Gyeongnam Agricultural Research and Extension Services, Jinju, South Korea. DNCB was procured from Sigma-Aldrich (St. Louis, MO, USA) and olive oil was obtained from a supermarket (South Korea). Antibodies against glyceraldehyde 3-phosphate dehydrogenase (GAPDH; sc-365062), TRAF6 (sc-8409), TNF-α (sc-52746), cyclooxygenase-2 (COX2; sc-376861) and inducible nitric oxide synthase (iNOS; sc-7271) were procured from Santa Cruz Biotechnology (Santa Cruz, CA, USA). Antibodies against MyD88 (4283S), IRAK4 (4363T), *p*-ERK (4377S), ERK (4695S), *p*-JNK (4668S), JNK (9258S), *p*-p38 (9215S) and *p*-38 (9212S) were obtained from Cell Signaling Technology (Danvers, MA, USA). 

### 2.2. Preparation of Protein Hydrolysates

Different molecular fractions, namely U-1 (<1 kDa), U-2 (1–10 kDa), U-3 (10–100 kDa), and U-4 (>100 kDa), of TMTH were collected as described in our previous study [13]. Briefly, lyophilized *T. molitor* larvae were ground to obtain a fine powder and then hydrolyzed with trypsin (1% (*w*/*w*) for 6 h at 38 °C (pH 7). The enzymes were inactivated by exposure to direct heating (100 °C) for 10 min. The enzyme-inactivated mixture was then centrifuged (Optima XE100 ultracentrifuge, Beckman Coulter, Miami, FL, USA) at 2000 rpm to obtain the supernatant. The supernatant was vacuum-filtered, followed by ultrafiltration (UF, Ultracel PL-10, Millipore, Billerica, MA, USA) with Amicon^®®^ Stirred Cells (Millipore, Billerica, MA, USA) with membrane cut-offs of 100 kDa, 10 kDa, and 1 kDa in a nitrogen atmosphere at 5 bar pressure. Different molecular fractions of proteins were thus obtained. Lyophilized molecular fractions were stored at −20 °C until further analysis.

### 2.3. Cell Culture and Treatment

Dulbecco’s modified Eagle’s medium supplemented with 10% fetal bovine serum (Hyclone, Logan, UT, USA) and 1% penicillin–streptomycin antibiotic (GIBCO, Grand Island, NY, USA) was used to culture HaCaT cells (Korean Cell Bank, Seoul, Korea). The cells were cultured in a humidified incubator maintained at 37 °C with 5% CO_2_ supply. After 24 h of incubation when the cell density reached 1 × 10^5^ cells/mL, the cells were treated with 200 µg/mL of three protein hydrolysates from *T. molitor* larvae: alkaline protease (TMAH), neutral protease (TMPH), and trypsin (TMTH) hydrolysates. After 30 min, HaCaT cells were treated with TNF-α and interferon (IFN)-γ at 10 ng/mL concentration for 6 h to induce inflammation. The same procedure was performed with four different molecular fractions of TMTH: U-1 (<1 kDa), U-2 (1–10 kDa), U-3 (10–100 kDa), and U-4 (>100 kDa), at 100 µg/mL. 

### 2.4. Reverse Transcription-Quantitative Polymerase Chain Reaction (RT-qPCR)

TRIzol reagent (Sigma-Aldrich, St. Louis, MO, USA) was used to extract RNA from HaCaT cells following the manufacturer’s instructions. RNA concentration and purity were measured using a nucleic acid protein concentration meter (PhileKorea, South Korea). The experiment was conducted according to the method described in a previous study [14]. Relative mRNA expression was calculated as follows: relative gene expression = 2^−(ΔCt of target gene–ΔCt of GAPDH)^, where Ct is the threshold cycle value, and GAPDH was used as an internal reference gene. The primer sequences used in this study are listed in Table 1.

### 2.5. Animal Model 

In this study, we used a black skin mouse model (C57BL/6) recommended for immunological studies. This is among the most commonly used mouse models for AD, as this strain has been reported to develop more severe skin contact irritation than BALB/C mice [15]. Eight-week-old male C57BL/6 mice (25 ± 2 g) were purchased from Nara Biotech Co., Ltd. (Pyeongtaek, Korea) and housed in clear plastic cages bedded with aspen chips. The cages were kept under pathogen-free conditions in a specific control room maintained at 20–21 °C with 40–45% relative humidity under a 12/12 h dark/light cycle. A standard mouse diet and tap water were provided ad libitum prior to the treatment. The study was approved by the Dong-A University Animal Care and Use Committee (DIACUC-217).

After two weeks of acclimatization, C57BL/6 mice were randomly allocated into five treatment groups (*n* = 5 per group): control group (CON), DNCB-treated group, TMTH (1 and 10 mg/kg, oral administration) with DNCB-treated group, and prednisolone (Youhanyanghaeng, Seoul, South Korea; 3 mg/kg) with DNCB-treated group (positive group). Dorsal hair was completely removed (approximately 4 × 3 cm) using an electric clipper one day before sensitization with DNCB. Subsequently, 200 μL of 1% DNCB was applied to the dorsal skin four days before oral administration and on day 1 for AD sensitization. An acetone: olive oil mixture (3:1, *v*/*v*) was used to prepare 1% DNCB. AD-like skin lesions were induced by applying 200 μL of 1% DNCB solution to the dorsal skin at two-day intervals for three weeks. Upon sensitization, sterile distilled water preparations of TMTH and prednisolone were administered orally (100 μL) daily for three weeks. Sterile distilled water (100 μL) was orally administered to the DNCB group mice. After three weeks of treatment, the mice were euthanized, and the respective samples were collected. The experimental design used in this study is summarized in Figure 1. 

### 2.6. Western Blot Analysis

Proteins extracted from the dorsal skin were analyzed by Western blotting. Lysis buffer containing protease inhibitors (Roche, Mannheim, Germany) was used to homogenize the skin tissues, followed by centrifugation at 4 °C and 15,928× *g* for 20 min to separate the supernatant. The bicinchoninic acid assay was used to determine the protein concentration of each sample. Proteins were denatured and separated using sodium dodecyl sulfate (SDS) buffer and 10% SDS-polyacrylamide gel electrophoresis, respectively. A Mini Trans-Blot cell (Bio-Rad, CA, USA) was used to transfer the proteins onto a nitrocellulose membrane. The protein-transferred membranes were blocked with 5% skim milk for 1 h at room temperature (25 ± 2 °C) before incubation at 4 °C overnight with primary antibodies, including GAPDH, MyD88, IRAK4, TRAF6, *p*-ERK, ERK, *p*-JNK, JNK, *p*-p38, *p*-38, TNF-α, iNOS, and COX2. All antibodies were used at a 1:1000 ratio. After washing the membranes three times with Tris-buffered saline + Tween 20 (TBST), the membranes were incubated with the corresponding secondary antibodies for 1 h at room temperature. Following three washes with TBST and processing with a horseradish peroxidase substrate (Advansta Inc., San Jose, CA, USA), the chemiluminescent membrane was imaged using an Azure c300 imaging system (Azure Biosystems, Dublin, CA, USA). 

### 2.7. Skin Histological Analysis of DNCB-Induced Mice 

Dorsal skin tissues were fixed in 10% formaldehyde for 24 h. The tissues were embedded in paraffin before sectioning into blocks of 4-µm thickness. Hematoxylin and eosin (H&E) and toluidine blue were used to stain tissue sections to determine the tissue architecture and degree of mast cell infiltration. Stained tissue sections were observed under a Leica DMi8 Research Inverted Phase microscope (Leica, Wetzlar, Germany) at ×100 and ×200 magnification. 

### 2.8. Determination of Serum IgE and IgG2a Levels in DNCB-Induced Mice

Serum was separated by centrifugation of mouse blood samples at 4 °C and 15,928× *g* for 20 min. The separated serum was stored at −80 °C until further analysis. Serum IgE and IgG2a levels were determined using a mouse IgE and IgG2a enzyme-linked immunosorbent assay kit (Bethyl Laboratories, Inc., Montgomery, TX, USA) according to the manufacturer’s instructions. 

### 2.9. Immunohistochemistry Analysis of DNCB-Induced Mice Skin

Mouse skin tissue specimens were fixed in formalin and embedded in paraffin. The fixed sections were then sectioned into 4-µm thick blocks. Following deparaffinization and rehydration, antigen retrieval was performed using sodium citrate buffer (10 mM sodium citrate, 0.05% Tween 20, pH 6.0). After blocking non-specific binding with 5% goat serum, tissue sections were incubated at 4 °C overnight with primary antibodies (anti-NF-κB, 1:500). After washing with TBST, the tissue sections were incubated with secondary antibodies at room temperature (25 ± 2 °C) for 2 h. After washing again with TBST, 3,3′-diaminobenzidine substrate solution was added to the tissue sections. After washing with distilled water, the sections were treated with hematoxylin. The dehydrated tissue sections were mounted and observed at ×400 magnification using a Leica DMi8 Research Inverted Phase microscope (Leica). 

### 2.10. Statistical Analysis

All quantitative data from the current study were statistically analyzed using GraphPad Prism 7.0 software (GraphPad Software, San Diego, CA, USA). The results are expressed as mean ± standard deviation of at least three replicates. Data were analyzed using a one-way analysis of variance and Tukey’s multiple comparison test. Statistical significance was set at *p* < 0.05.

## 3. Results

### 3.1. Effect of TMTH on TNF-α-Stimulated HaCaT Cell 

To screen for the effect of *T. molitor* protein hydrolysate on proinflammatory cytokine expression in HaCaT cells, three protein hydrolysates (TMAH, TMPH, and TMTH) were used. TNF-α and IFN-γ were used to stimulate inflammation in HaCaT cells to represent inflammatory skin diseases in an in vitro model. Our results showed that TNF-α overexpression was downregulated by treatment with 200 µg/mL of TMTH. However, TMAH and TMPH did not downregulate TNF-α expression at 200 µg/mL (Figure 2A). Figure 2B shows the expression of TNF-α after treatment with four different molecular fractions of TMTH obtained by ultrafiltration: U-1 (<1 kDa), U-2 (1–10 kDa), U-3 (10–100 kDa), and U-4 (>100 kDa). The results revealed that U-2, U-3, and U-4 significantly downregulated the expression of TNF-α. Levels of IL-6 and IL-8 expression exhibited a similar trend; overexpression of IL-6 and IL-8 was effectively downregulated by U-2, U-3, and U-4. Moreover, U-1 downregulated the expression of IL-6 and IL-8. Considering the promising effects of U-2 on TNF-α-stimulated HaCaT cells, we selected U-2 for further studies in DNCB-induced mice.

### 3.2. Effects of TMTH on DNCB-Induced AD-like Lesions in the Skin of C57BL/6 Mice

In the present study, DNCB was used to induce AD-like lesions in the skin of C57BL/6 mice to investigate the therapeutic efficacy of TMTH in AD. The AD group exhibited significant physical signs of AD, such as hemorrhage, erythema, and dryness, compared with the control group. Moreover, in comparison with the control group, the mice in the AD group exhibited conspicuous epidermal thickening, drying shrinkage, and crust on their backs. However, oral administration of TMTH and prednisolone for three weeks significantly decreased the DNCB-induced AD (Figure 3A). Thus, TMTH and prednisolone alleviated AD clinical symptoms. Figure 3B shows the body weights of the mice treated with TMTH and the positive control. As shown in the figure, the body weight of the mice did not exhibit noticeable differences between the different treatment groups: control group and the AD, TMTH (10 mg/kg), and prednisolone groups. Figure 3C shows the dermatitis scores of each treatment group. As shown in the figure, the dermatitis score increased rapidly during the first week. The AD group showed the highest score, while TMTH treatment gradually decreased the dermatitis score. This decrease was more pronounced at high concentrations of TMTH, similar to the effect of prednisolone.

### 3.3. Effects of TMTH on Immune Organs of AD Mice

The current study evaluated the anti-AD properties of TMTH by measuring the weight and size of the spleens and lymph nodes (LNs) of DNCB-treated AD mice. Figure 4A shows the sizes of the LNs and spleens in each group. The LNs and spleens in the AD group were larger than those in the control group. Moreover, size of LNs and spleens of the TMTH-L and TMTH-H groups were less significant than those in the AD group. Similarly, the LNs and spleens of the prednisolone-treated group were smaller than those of the AD group were. Figure 4B–D shows the weights of the LNs and spleen of the different treatment groups. The AD group exhibited the highest spleen weight, and treatment with TMTH reduced spleen weight. Prednisolone, TMTH-L, and TMTH-H showed significantly (*p* < 0.05) low spleen weight compared to the AD group. Interestingly, there was no significant difference between the spleen weights of the control and TMTH-treated groups. The weight of the axillary LNs followed the same pattern. The AD group showed the highest axillary LN weight, whereas the control group showed the lowest axillary LN weight. Moreover, there was no significant difference between the axillary LN weights in the TMTH-H and prednisolone groups. Similarly, the AD group showed the highest inguinal LN weight, whereas the control group exhibited the lowest weight. Moreover, treatment with TMTH and prednisolone reduced the weight of the inguinal LNs. The spleen weight/body weight ratio, axillary LN weight/body weight ratio, and inguinal LN weight/body weight ratio of the different treatment groups are shown in Figure 4E–G. The AD group showed the highest spleen weight/body weight ratio, whereas the control group showed the lowest ratio. Spleen weight/body weight ratio of the control and TMTH-H groups was insignificant (*p* > 0.05). The ratios of axillary LN weight/body weight and inguinal LN weight/body weight also showed a similar pattern, and the AD group exhibited the highest ratio, whereas the control group exhibited the lowest ratio. Moreover, TMTH and prednisolone treatment decreased both axillary LN weight/body weight and inguinal LN weight/body weight ratios. 

### 3.4. Effect of TMTH on TLR-Mediated MyD88-Dependent MAPK Signaling Pathway in DNCB-Induced Mice

The current study investigated the molecular mechanisms involved in TMTH and determined the expression levels of MyD88, IRAK4, and TRAF6 proteins associated with the TLR-mediated MyD88-dependent signaling pathway. The results revealed that the expression levels of MyD88, IRAK4, and TRAF6 were higher in the DNCB-treated group than those in the control group (Figure 5A). Treatment with TMTH decreased the expression levels of MyD88, IRAK4, and TRAF6. However, prednisolone treatment was more effective in reducing the expression of MyD88 and TRAF6 protein, where prednisolone decreased the expression of MyD88 protein, similar to that in the control group, and there was no significant difference (*p* > 0.05) between the MyD88 protein expression of control group and prednisolone group. In contrast, the IRAK4 and TRAF6 protein expression levels in the prednisolone-treated group were similar to those in the TMTH-treated group. There was no significant difference (*p* > 0.05) between the TRAF6 and IRAK4 protein expression of the TMTH-H group and prednisolone group. 

Western blotting was performed to measure the effect of TMTH on mitogen-activated protein kinases in skin tissue. The expression levels of *p*-p38, *p*-ERK, and *p*-JNK were significantly higher in the DNCB-treated group than in the control group (Figure 5B). However, TMTH treatment ameliorated the DNCB-induced increase in *p*-p38, *p*-ERK, and *p*-JNK expression. The results further revealed that TMTH treatment significantly (*p* < 0.05) decreased the protein expression of *p*-ERK/ERK and *p*-p38/*p*-38 compared to those in the DNCB-treated group. In contrast, TMTH treatment decreased the expression of *p*-JNK/JNK compared to that in the AD group. Interestingly, there was no significant difference in the expression of *p*-ERK/ERK and *p*-p38/*p*-38 between the TMTH-H and prednisolone groups. 

The current study further measured the effect of TMTH on the expression levels of inflammatory proteins, including tumor necrosis factor-α (TNF-α), inducible nitric oxide (iNOS), and cyclooxygenase (COX-2). Results revealed that the DNCB treatment increased the expression of TNF-α, iNOS, and COX-2 in mice (Figure 5C). TMTH treatment decreased the DNCB-induced expression of TNF-α, iNOS, and COX-2. TMTH exerted a dose-dependent effect on TNF-α expression. Interestingly, TNF-α expression in the high concentration of TMTH group and prednisolone group was significant (*p* > 0.05). The expression of iNOS followed a similar pattern, where TMTH treatment effectively decreased the expression of iNOS. Furthermore, iNOS expression of the control group and TMTH-treated group was insignificant (*p* > 0.05).

### 3.5. Effects of TMTH on Mast Cell Infiltration and Dermal and Epidermal Thickness 

A DNCB-treated mouse model was used to analyze the effects of TMTH on AD-like lesions in C57BL/6 mice. Mouse skin histomorphology was performed using H&E staining to examine the effects of TMTH on the pathological tissues of mouse skin. The results showed that the dermal and epidermal tissues were remarkably thickened by DNCB treatment, whereas TMTH treatment gradually decreased the dermal and epidermal thickness (Figure 6A,B). The dermal thickness at low and high doses of TMTH was not significantly different (*p* > 0.05), whereas the epidermal thickness decreased in a concentration-dependent manner following oral administration of TMTH in DNCB-induced mice (Figure 6C,D). Interestingly, the epidermal thickness in the TMTH-H and prednisolone groups was insignificant.

Toluidine blue staining was performed on the dorsal skin of DNCB-induced mice to measure the infiltration of mast cells, as mast cell infiltration is one of the major characteristic features of inflamed skin [16]. As shown in Figure 6E, DNCB treatment increased the mast cell count, whereas TMTH treatment decreased mast cell infiltration. Thus, TMTH treatment effectively controlled infiltration of mast cells in DNCB-treated mice. Figure 6F shows a dose-dependent decrease in mast cell infiltration in the skin of TMTH-treated C57BL/6 mice. TMTH treatment significantly decreased mast cell infiltration compared to the AD group. Moreover, the mast cell counts of the control- and prednisolone-treated group were insignificant (*p* > 0.05). Interestingly, the mast cell counts in the TMTH-treated group and prednisolone-treated group were insignificant (*p* > 0.05). 

### 3.6. Effects of TMTH on Serum IgE and IgG2a Levels in DNCB-Induced AD Mice

Repeated exposure to DNCB led to elevated levels of IgE and IgG2a. As IgE hypersecretion has been identified as the primary etiology of AD [17], the effects of TMTH on serum IgE and IgG2a levels were also analyzed in this study. DNCB markedly increased IgE (Figure 7A) and IgG2a (Figure 7B) levels. However, TMTH treatment significantly decreased the serum IgE and IgG2a levels in a concentration-dependent manner. However, only IgE showed a significant difference between the low and high concentrations of TMTH. The results demonstrated that TMTH could effectively decrease the elevated levels of serum IgE and IgG2a in DNCB-induced C57BL/6 mice, thereby downregulating histopathological and immunological characteristics.

### 3.7. NF-κB Expression Levels in Dorsal Skin of DNCB-Induced Mice

Immunohistochemistry was used to investigate the effects of TMTH on the expression levels of nuclear NF-κB in the dorsal skin of DNCB-treated mice. According to the immunohistochemical analysis, there was a notable difference between the AD and TMTH-treated groups (Figure 8A). The AD group exhibited relatively high NF-κB expression in the nuclei compared to the control group, as indicated by the red arrows. As revealed by the immunohistochemistry results, TMTH treatment led to weak positive staining in the nuclei. Thus, NF-κB expression was markedly decreased by TMTH treatment, and this decrease was more pronounced at high concentrations of TMTH. As shown in Figure 8B, the mean optical density of NF-κB in the control group and TMTH-H group was insignificant (*p* > 0.05). Moreover, there was no significant (*p* > 0.05) difference between the mean optical density of NF-κB expression in TMTH-L and the positive control prednisolone.

## 4. Discussion

AD is a chronic inflammatory skin disease characterized by symptoms such as erythema, swelling, and cracked skin. Genetic and epigenetic factors, environmental and immunological interactions, and epidermal barrier defects are the major causes of AD. Current treatments for AD include anti-inflammatory drugs, antihistamines, and topical therapy with emollients [18,19]. However, the usage of these drugs is linked with several adverse effects. Therefore, exploration of natural sources for the treatment of this chronic inflammatory disease has gained much attention in recent years. Thus, the current study assessed the anti-AD effects of TMTH in DNCB-induced C57BL/6 mice by examining TLRs and their adapter proteins, as well as the downstream intracellular signaling pathways, such as the MAPK pathway.

The findings of the current study demonstrate that oral administration of TMTH can alleviate the inflammatory manifestations in DNCB-induced AD mice. This study further explored the anti-inflammatory mechanism of TMTH. Several studies have reported the role of TLRs in the progression and aggravation of AD. The adapter protein MyD88 binds to TLR2 and stimulates its receptor, thereby activating IRAK-4 kinase and leading to IRAK-1 phosphorylation. Activated IRAK-1 is released into the cytoplasm in association with TRAF6 factor and ultimately leads to the activation of IKK-IκB kinase and MAPK. This activation of the downstream signaling cascade leads to the activation of NF-κB releasing proinflammatory cytokines [18]. Thus, MyD88, IRAK4, and TRAF6 play major roles in the initiation and progression of AD, and our findings demonstrated the lower expression of MyD88, IRAK4, and TRAF6 with TMTH treatment in DNCB-treated AD mice (Figure 4A). Kordulewska et al. also highlighted the importance of orchestrating TRAF6 signaling via TLRs and IL-17 receptors in the development and progression of IL-17-mediated immunity in the skin [18]. Thus, the ability of TMTH to interact with this signaling pathway is important for the formulation of potential drugs to treat AD.

TLR stimulation activates the MAPK signaling pathway, which is associated with several cellular functions. Thus, numerous studies have proven the role of different constituents of the MAPK pathway, namely ERK, JNK, and p38, in the pathogenesis of AD, which influences cell proliferation, differentiation, and inflammatory response [20,21]. In this study, DNCB was used as a skin stimulation agent that exhibits a function similar to that of hapten. Hapten is a small molecule that can easily penetrate the epidermis and bind to tissue proteins, inducing an immune response and subsequently leading to AD development [22]. Thus, DNCB application activated the MAPK pathway, which is one of the major pathways involved in dermal inflammation. The findings of the current study revealed that DNCB application increased the expression levels of *p*-ERK, *p*-JNK, and *p*-p38 compared to those in the control group (Figure 4B). In contrast, oral administration of TMTH decreased the expression of *p*-JNK, *p*-ERK, and *p*-p38 compared with that in the AD group. Thus, TMTH inhibited the phosphorylation of the MAPK constituents. The findings of the present study followed the pattern of a previous study in which *Peucedanum japonicum* Thunberg, an herbal medicine, suppressed the phosphorylation of ERK [22]. Moreover, a study by Liu et al. also reported similar results; Angelica Yinzi, a classic ancient prescription, blocks ERK, JNK, p38 phosphorylation [16]. MAPK activation has been identified as one of the major causes of AD because MAPK phosphorylation promotes inflammatory cell generation. Studies have shown that suppression of MAPK phosphorylation inhibits NF-κB signaling pathway activation, thereby attenuating the inflammatory responses of the skin [23,24]. Owing to the involvement of MAPK in the synthesis of inflammatory mediators, they have been identified as potential targets for anti-inflammatory therapeutics. As TMTH effectively suppresses the MAPK signaling pathway.

The skin provides the first line of defense against allergens of external origin. Thus, skin barrier homeostasis has been identified as a major contributing factor to AD [25]. Thus, the present study analyzed the histopathological characteristics of DNCB-induced C57BL/6 mice. H&E staining revealed that TMTH treatment effectively decreased the thickness of the epidermis and dermis in a concentration-dependent manner (Figure 5A). Toluidine blue staining was used to measure mast cell infiltration. Mast cells contribute to allergic reactions through the production and secretion of proinflammatory mediators such as histamine, chemokines, cytokines, and growth factors. Moreover, mast cells actively contribute to modulating the function of naive T cells, thereby increasing T-cell activation [2]. The current study revealed that TMTH administration significantly decreased mast cell infiltration in DNCB-treated mice (Figure 5E). An increasing number of mast cells have been identified as a major characteristic of patients with AD. Mast cells are known to contribute to AD pathogenesis [26]. As revealed by the histopathological results, TMTH treatment decreased the infiltration of mast cells and thickness of the epidermis and dermis, thereby regulating AD-like skin lesion formation. The findings of the present study are consistent with those of previous studies in which glabridin, a major hydrophobic constituent found in the roots and rhizomes of *Glycyrrhiza glabra* L, effectively downregulated mast cell infiltration and decreased epidermal and dermal thickness [1]. Moreover, similar results were reported with the treatment of DNCB-induced mice with Angelica Yinzi, a classic ancient prescription [16]. Lee et al. also reported decreased mast cell infiltration and epidermal and dermal thickness following treatment of DNCB-treated mice with *Centella asiatica* extract [27].

Elevated NF-κB levels are a major characteristic of inflammation. Lipopolysaccharide signaling, T-cell receptors, and TNF primarily contribute to the stimulation of NF-κB. Upon activation of NF-κB, protein translocation to the nucleus occurs. This regulates the expression of inflammatory factors, which is a well-known hallmark of the inflammatory response [28]. In this study, we used immunohistochemistry to investigate the effect of TMTH on NF-κB signaling. Our results showed that NF-κB p65-regulated local inflammation markedly increased in the DNCB-treated group. The findings of the current study are consistent with those of previous studies in which improved NF-κB p65 signaling was observed in AD mice [28,29].

IgE contributes to the development of AD by mediating mast cell activation [30]. Mast cells are activated by the binding of IgE to high-affinity Fcε receptor (FcεRI) on the surface of mast cells. Antigen IgE cross-linking triggers FcεRI-mediated mast cell activation, leading to the expression of proinflammatory mediators. Elevated IgE levels and an increased number of mast cells have been observed in the majority of patients with AD. IgE is well known for its effects on the pathogenesis associated with allergic inflammatory diseases, as elevated serum levels of IgE mediate the vital attributes of AD [31]. Therefore, IgE is one of the most crucial therapeutic targets for AD. The results of our study revealed that the AD group exhibited elevated levels of IgE compared to the control group, as shown in Figure 6A, and that TMTH treatment decreased serum IgE levels in DNCB-induced mice in a concentration-dependent manner. Moreover, when high levels of TMTH (10 mg/kg) were administered, the serum IgE level was similar to that of the positive control prednisolone. In the current study, prednisolone was administered at a dose of 3 mg/kg, considering similar studies carried out with prednisolone [32,33,34,35,36]. Thus, TMTH could be effectively utilized as a potential remedy for AD management.

## 5. Conclusions

This study investigated the immunomodulatory and anti-inflammatory effects of TMTH in mice with DNCB-induced AD. The results of this study revealed that TMTH decreased DNCB-induced elevation of IgE and IgG2a levels, and decreased epidermal thickness. TMTH targeted the TLR-mediated MyD88-dependent MAPK signaling pathway to suppress the expression of MyD88, IRAK4, and TRAF6. Moreover, TMTH suppressed MAPKs phosphorylation and inhibited activation of the NF-κB signaling pathway, thereby attenuating inflammatory responses in the skin. These findings confirmed the therapeutic effects of TMTH for potential applications in AD treatment. Further studies are needed to analyze other active compounds of TMTH and determine their potential effects on AD.

## Figures and Tables

**Figure 1 nutrients-15-00093-f001:**
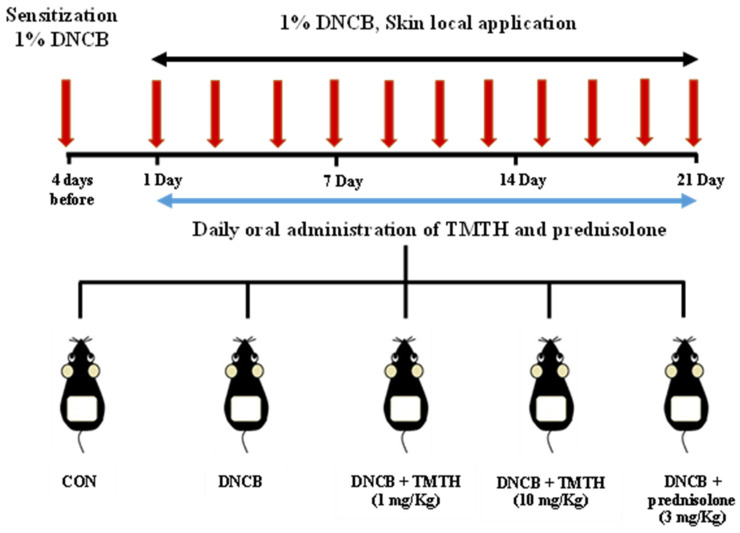
Sensitization of C57BL/6 mice was performed using 1% 1-chloro-2,4-dinitrobenzene (DNCB) four days before oral administration and on day 1. Subsequently, mice were challenged with 1% DNCB at two-day intervals for three weeks. *Tenebrio molitor* trypsin hydrolysate (TMTH; 1 and 10 mg/kg) and prednisolone (3 mg/kg) were orally administered daily for three weeks to treat atopic dermatitis (AD)-like lesions.

**Figure 2 nutrients-15-00093-f002:**
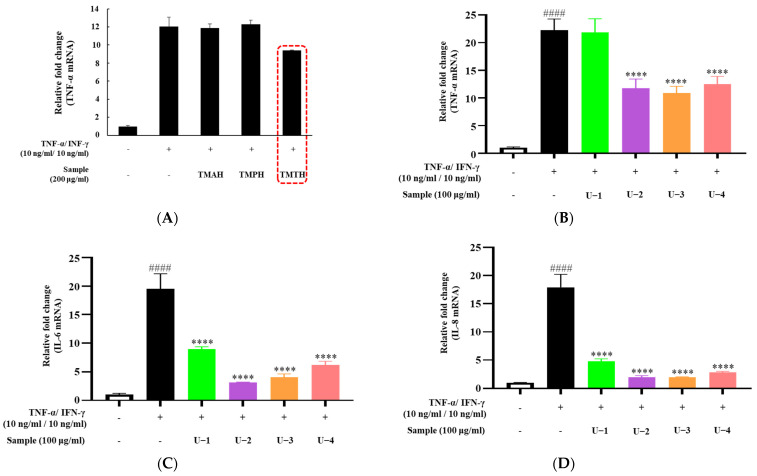
Effects of different protein hydrolysates of *Tenebrio molitor* larvae and different molecular fractions of *T. molitor* trypsin hydrolysate (TMTH) on the expression levels of cytokines in HaCaT cells. The cells were pretreated with three protein hydrolysates of *T. molitor* larvae: alkaline protease (TMAH), neutral protease (TMPH), and trypsin (TMTH) hydrolysates, and four molecular fractions of TMTH: U-1 (<1 kDa), U-2 (1–10 kDa), U-3 (10–100 kDa), and U-4 (>100 kDa) at 200 and 100 µg/mL concentrations. Subsequently, cells were induced with interferon (IFN)-γ (10 ng/mL)/tumor necrosis factor (TNF)-α (10 ng/mL) for 6 h. (**A**) TNF-α mRNA expression of TMAH-, TMPH-, and TMTH-treated HaCaT cells. (**B**) TNF-α, (**C**) interleukin (IL)-6, and (**D**) IL-8 mRNA expression of U-1, U-2, U-3, and U-4 treated HaCaT cells. Each bar represents the mean of three replicates. Data are expressed as the mean ± standard deviation (SD) of three replicates. #### *p* < 0.0001 vs. untreated control; **** *p* < 0.0001 vs. TNF-α/IFN-γ group. IL: interleukin; TNF: tumor necrosis factor; IFN: interferon.

**Figure 3 nutrients-15-00093-f003:**
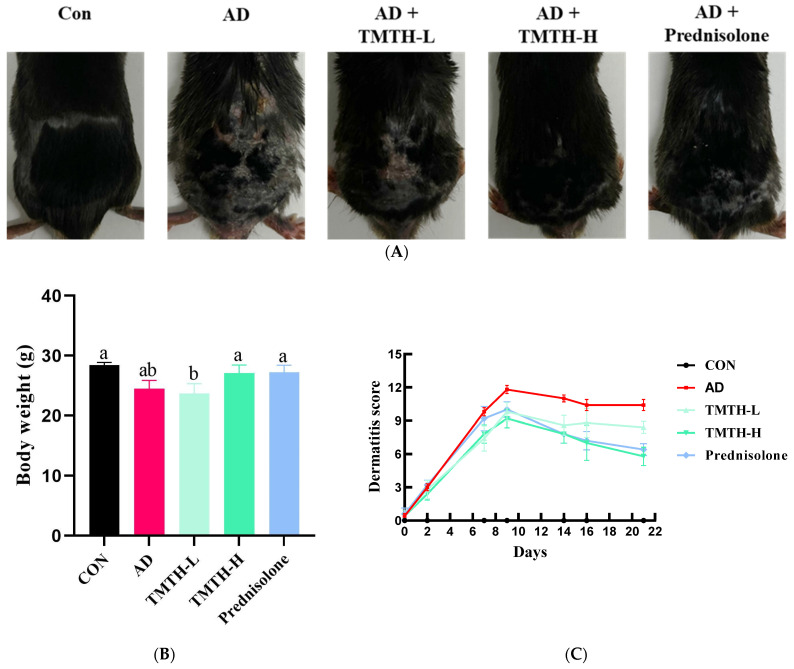
Effects of *Tenebrio molitor* trypsin hydrolysate (TMTH) on DNCB-induced AD-like lesions in the skin of C57BL/6 mice. (**A**) Clinical features of the control group, AD group, and DNCB-induced mice treated with TMTH and prednisolone group after three weeks. (**B**) Average body weight of control group, AD group, and DNCB-induced mice treated with TMTH and prednisolone group after three weeks. (**C**) Dermatitis score of the control group, AD group, and DNCB-induced mice treated with TMTH and prednisolone group after three weeks. The significant values are indicated based on the Tukey’s multiple comparisons test (*p* < 0.05) and results are expressed as the mean ± standard deviation (SD). Different letters indicate statistically significant differences (*p* ≤ 0.05) among treatments. CON: control group; AD: DNCB-treated group; TMTH-L: mice challenged with 1% 2,4-dinitrochlorobenzene (DNCB).

**Figure 4 nutrients-15-00093-f004:**
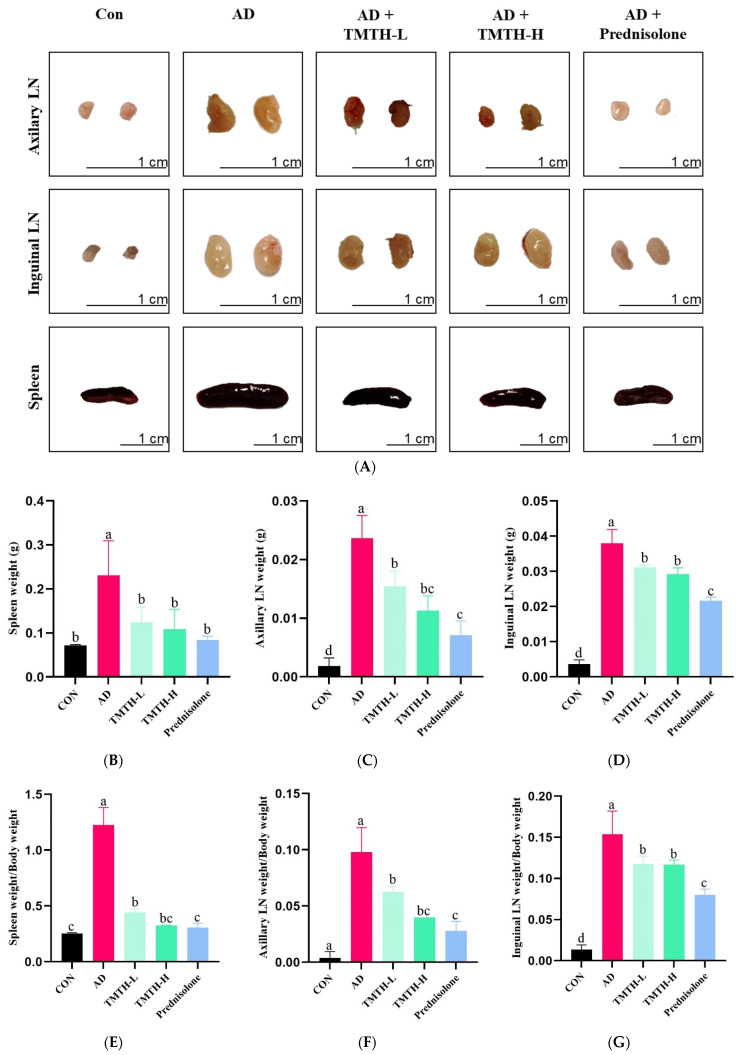
Effects of *Tenebrio molitor* trypsin hydrolysate (TMTH) on immune organs of 1-chloro-2,4-dinitrobenzene (DNCB)-induced mice. (**A**) Photographs of axillary lymph nodes (LNs), inguinal LNs, and spleens of the control group, AD group, and DNCB-induced mice treated with TMTH and prednisolone group after three weeks. Weight of spleen (**B**), Axillary LNs (**C**) and inguinal LNs (**D**) of each treatment group. (**E**) Spleen weight/body weight ratio, (**F**) axillary LN weight/body weight ratio, and (**G**) inguinal LN weight/body weight ratio of each treatment group. CON: control group; AD: DNCB-treated group; TMTH-L: mice challenged with 1% 2,4-dinitrochlorobenzene (DNCB) treated with 1 mg/kg TMTH; TMTH-H; mice challenged with 1% 2,4-dinitrochlorobenzene (DNCB) treated with 10 mg/kg TMTH; Prednisolone: mice challenged with 1% DNCB treated with 3 mg/kg prednisolone. Different letters indicate statistically significant differences (*p* ≤ 0.05) among treatments.

**Figure 5 nutrients-15-00093-f005:**
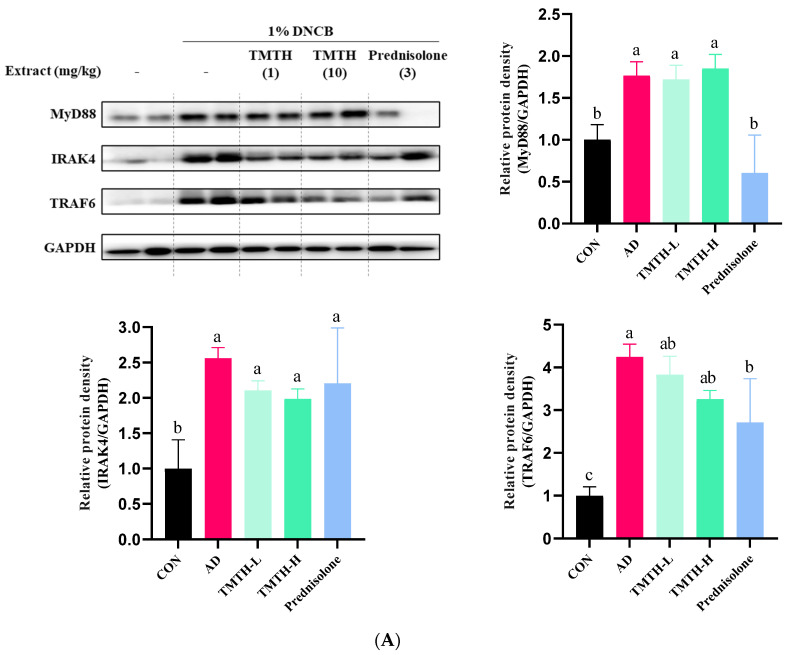
Effects of *Tenebrio molitor* trypsin hydrolysate (TMTH) on the protein expression levels of inflamed dorsal skin. (**A**) Effect of TMTH on myeloid differentiation primary response 88 (MyD88)-dependent Toll-like receptor (TLR) signaling pathway. (**B**) Effect of TMTH on mitogen-activated protein kinase (MAPK) signaling pathway. (**C**) Expression levels of inflammatory proteins, tumor necrosis factor-α (TNF-α), inducible nitric oxide synthase (iNOS), and cyclooxygenase (COX-2). The relative protein densities were presented as a ratio to those of GAPDH. The data are expressed as the mean ± standard deviation (SD) of three replicates. The significant values are denoted by different letters as per the results of the Tukey’s multiple comparisons test (*p* < 0.05). CON: control group; AD: DNCB-treated group; TMTH-L: mice challenged with 1% 2,4-dinitrochlorobenzene (DNCB) treated with 1 mg/kg TMTH; TMTH-H; mice challenged with 1% DNCB treated with 10 mg/kg TMTH; Prednisolone: mice challenged with 1% DNCB treated with 3 mg/kg prednisolone.

**Figure 6 nutrients-15-00093-f006:**
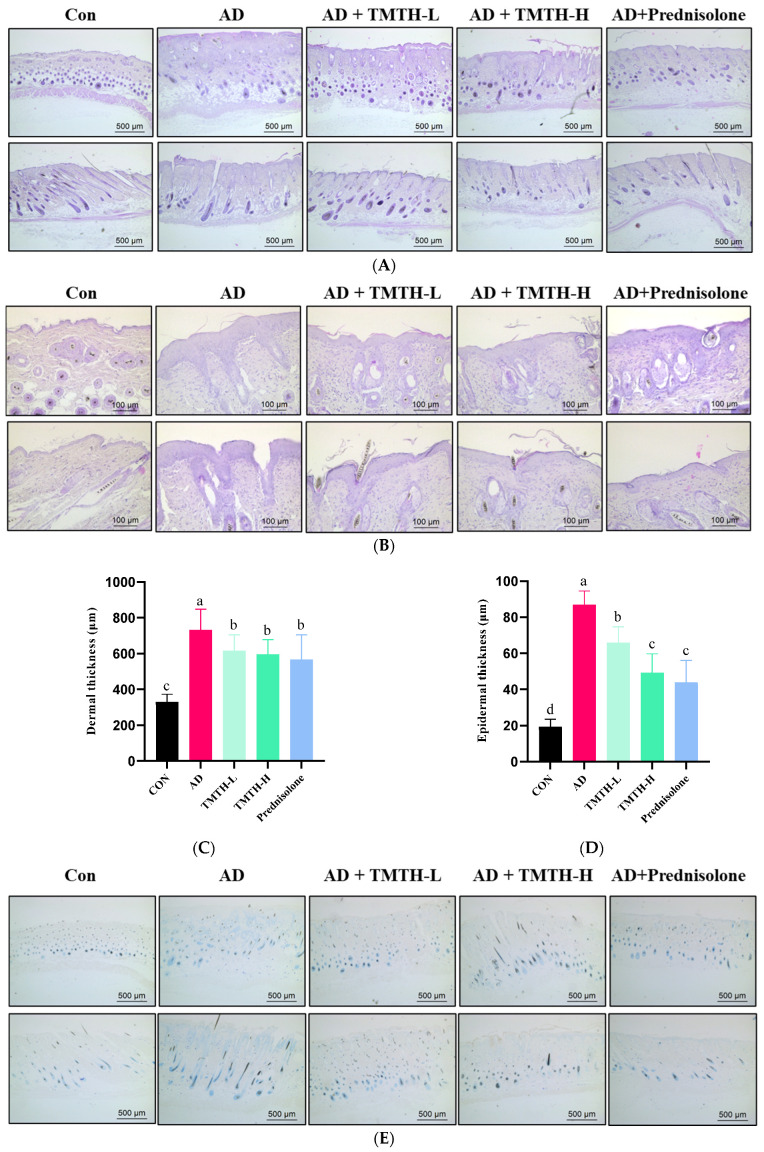
Effects of *Tenebrio molitor* trypsin hydrolysate (TMTH) on the histopathological characteristics of DNCB-induced mice. (**A**) H&E staining photographs of cross and longitudinal sections of dorsal skin at ×100 magnification (500 µm; Scale bar). (**B**) Hematoxylin and eosin (H&E) staining photographs of cross and longitudinal sections of dorsal skin at ×200 magnification (100 µm; Scale bar). (**C**) Dermal thickness of control group, AD group, and DNCB-induced mice administered TMTH and prednisolone group. (**D**) Epidermal thickness of control group, AD group, and DNCB-induced mice treated with TMTH and prednisolone group. (**E**) Toluidine blue staining photographs of dorsal skin of control group, AD group, and DNCB-induced mice treated with TMTH and prednisolone group at ×100 magnification (Scale bar, 500 µm). (**F**) Mast cell number of control group, AD group, and DNCB-induced mice treated with TMTH and prednisolone group. The significant values are expressed in different letters based on the Tukey’s multiple comparisons test results (*p* < 0.05). CON: control group; AD: DNCB-treated group; TMTH-L: mice challenged with 1% 2,4-dinitrochlorobenzene (DNCB) treated with 1 mg/kg TMTH; TMTH-H; mice challenged with 1% DNCB treated with 10 mg/kg TMTH; Prednisolone: mice challenged with 1% DNCB treated with 3 mg/kg prednisolone.

**Figure 7 nutrients-15-00093-f007:**
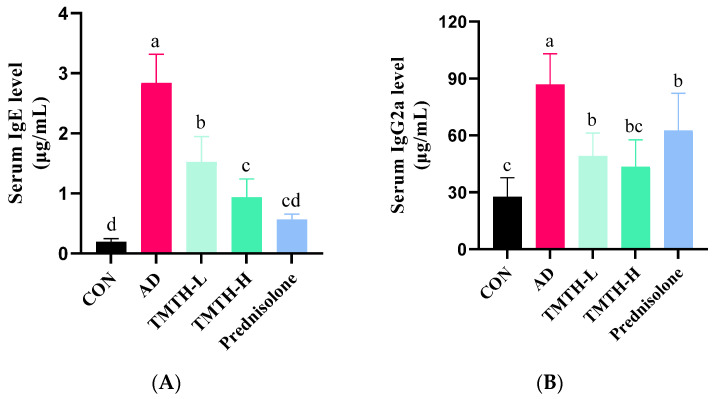
Serum IgE and IgG2a levels in DNCB-induced mice. (**A**) Serum IgE levels in the control group, AD group, and DNCB-induced mice treated with *Tenebrio molitor* trypsin hydrolysate (TMTH) and prednisolone group. (**B**) Serum IgG2a levels in the control group, AD group, and DNCB-induced mice treated with TMTH and prednisolone group. The data are expressed as the mean ± standard deviation (SD) of results of three replicates. The significant values are denoted by different letters based on the Tukey’s multiple comparisons test results (*p* < 0.05). CON: control group; AD: DNCB-treated group; TMTH-L: mice challenged with 1% 2,4-dinitrochlorobenzene (DNCB) treated with 1 mg/kg TMTH; TMTH-H; mice challenged with 1% DNCB treated with 10 mg/kg TMTH; Prednisolone: mice challenged with 1% DNCB treated with 3 mg/kg prednisolone.

**Figure 8 nutrients-15-00093-f008:**
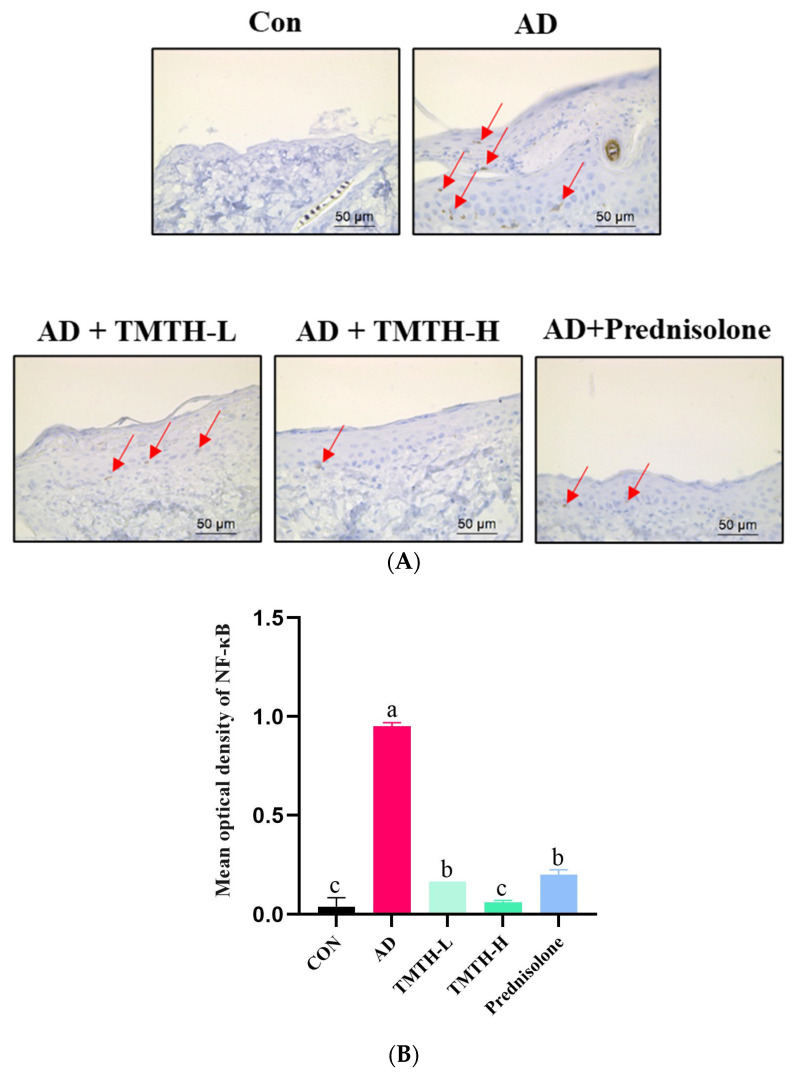
Immunohistochemical analysis of nuclear factor (NF)-κB expression levels in DNCB-treated AD mice treated with *Tenebrio molitor* trypsin hydrolysate (TMTH) and prednisolone. (**A**) NF-κB expression of dorsal skin tissue of DNCB-induced mice ×400 magnification (50 µm; Scale bar). (**B**) Mean optical density of NF-κB expression. ImageJ software was used to quantify the expression level of NF-κB. After selection of area of interest with positive expression, integrated optical density was calculated in relation to area of sum. CON: control group; AD: DNCB-treated group; TMTH-L: mice challenged with 1% 2,4-dinitrochlorobenzene (DNCB) treated with 1 mg/kg TMTH; TMTH-H; mice challenged with 1% DNCB treated with 10 mg/kg TMTH; Prednisolone: mice challenged with 1% DNCB treated with 3 mg/kg prednisolone.

**Table 1 nutrients-15-00093-t001:** Primer sequences of the genes used in the study.

Gene	Primer	Sequence
*GAPDH*	Forward	5′-CCCCTGGCCAAGGTCATCCATGACAACTTT-3′
	Reverse	5′-GGCCATGAGGTCCACCACCCTGTTGCTGTA-3′
*IL-6*	Forward	5′-AAAGAGGCACTGCCAGAAAA-3′
	Reverse	5′-ATCTGAGGTGCCCATGCTAC-3′
*IL-8*	Forward	5′-ACAGCAGAGCACACAAGCTT-3′
	Reverse	5′-CTGGCAACCCTACAACAGAC-3′

## Data Availability

All data generated in the current study are contained within this article.

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
