# Peer review of "Tenebrio molitor Larva Trypsin Hydrolysate Ameliorates Atopic Dermatitis in C57BL/6 Mice by Targeting the TLR-Mediated MyD88-Dependent MAPK Signaling Pathway"

_nutrients, 2022, doi:10.3390/nu15010093_

Round 1

Reviewer 1 Report

The main topic of the research paper was to check if Tenebrio molitor larva trypsin hydrolysate possessed anti-inflammatory properties in a model of atopic dermatitis (AD). The Authors applied in vitro cellular model to induce an inflammatory response by TNF-a/IFN-g and to verify if different fractions of T.molitor hydrolysate could modify the level of selected interleukins. Moreover, they utilized dinitrochlorobenzene-induced contact hypersensitivity in mice, which is an established model of AD. The research objectives and results are clearly presented. The obtained results about the anti-inflammatory properties of this substance and its comparison to standard steroid treatment are interesting. However, I have found are some issues to address and improve:

1. “Preparation of protein hydrolysates” section and the process of hydrolysis must be described in detail. As the authors wrote, the various functional properties of T.molitor protein hydrolysated depend on the method of extraction. In the given literature reference peptides were isolated from another insect. In the case of extracts obtained from natural sources, the exact method of obtaining them must be clearly described.

Also, section 2.9 needs more details (e.g. which method was used for antigen retrieval or which solution for blocking non-specific binding).

2. Line 91: Should it be TMTH?

Lines 93-97: it should be noted that these all are antibodies, the catalog numbers should be added here (in Materials) or in the “Western blotting” section.

Line 231: it states mistakenly that “TMAH and TMPH downregulated TNF-a expression at 200mg/mL”, while in Figure 2A it is not seen.

In Figure 3 and 4 it is not explained that different letters indicate statistical significance

Line 422-423: states that “Treatment with TMTH decreased the expression of MyD88, IRAK4, and TRAF6.” But I do not see it in the graph or in statistical significance indicated by different letters. Please be precise in describing the increase or the decrease.

Lines 426, 431, 452, 455, and 568: it should be p > 0.05 if there is no statistical difference

I have some doubts about Figure 5B – e.g. on the graph the results are presented as p38/p-p38 for each group. When I look at representative Western blot images, I see a rather opposite ratio. Was the GAPDH control included in your data on graphs? The same impression applies also to ERK/p-pERK and JNK/p-JNK presentations when comparing graphs with representative images.

3. Western blot for ERK shows two bands indicating ERK1 and ERK2. The authors present total values for both forms. Is it possible to calculate the expression level separately for ERK1 and ERK2? If yes, are there any differences? The same could be applied to pERK and JNK/pJNK.

4. Is it possible to quantify the differences in expression level in NF-kB in nuclei?

5. Have you got any data if the studied hydrolysate could have any anti-inflammatory properties if it is applied directly to the skin, what could be useful in AD treatment and mentioned in the Discussion?

Author Response

We thank you for taking out time to review our manuscript. Your comments and suggestions have been very useful in improving the quality of our manuscript. 

Reviewer 2 Report

Congratulations for the hard and detailed work.

I have some minor comments.

1. First of all, about AD: No one ever calls it ectopic (line 40). It should be deleted. In line 724: antibiotics are generally contrainticated in AD, especially p.os. Rarely, they are used (mainly topically), in cases of superinfections.

2. Can the authors tell some details about C57BL/6 mice? What are they and why were they chosen? Lab scientists may be familiar, but clinicians no.

3. Lines 230-232 are saying the same thing. All 3 downregulated TNF at 200μg. Is this wrong, or figure 2A is wrong? Or I don't understand something correctly?

4. In human patients (probably in mice also) inflammatory diseases like AD, psoriasis etc. are sometimes unpredictable. The same drug may work in a patient and not in another, or it may stop working in the same patient etc. So, sometimes, lab results like level of IgE, various cytokines etc. may not correlate with disease severity or course. Since the disease in mainly a skin inflammation, it would be helpful to mention some more findings from skin biopsies (spongiosis, infiltration etc.) if, of course, they are available.

5. Why was the dosage of 3mg for prednisolone chosen? Are there any "guidelines" for treatment of AD in mice, in the context of experiments like those presented here? In humans, 3mg/Kgr would be an enormous dose. If the same is true for mice, it may even be an unfair comparison for TMTH.

Thank you

Author Response

(The authors gave the same response as above.)
